# Machine Learning in Assessing the Performance of Hydrological Models

Evangelos Rozos [1,*], Panayiotis Dimitriadis [2] and Vasilis Bellos [3]

[1] Institute for Environmental Research & Sustainable Development, National Observatory of Athens, 15236 Athens Greece

[2] Department of Water Resources and Environmental Engineering, School of Civil Engineering, National Technical University of Athens, 15780 Athens, Greece; pandim@itia.ntua.gr

[3] Department of Environmental Engineering, School of Engineering, Democritus University of Thrace, 67100 Xanthi, Greece; vbellos@env.duth.gr

[*] Correspondence: erozos@noa.gr; Tel.: +30-210-810-9125

**Abstract:** Machine learning has been employed successfully as a tool virtually in every scientific and technological field. In hydrology, machine learning models first appeared as simple feed-forward networks that were used for short-term forecasting, and have evolved into complex models that can take into account even the static features of catchments, imitating the hydrological experience. Recent studies have found machine learning models to be robust and efficient, frequently outperforming the standard hydrological models (both conceptual and physically based). However, and despite some recent efforts, the results of the machine learning models require significant effort to interpret and derive inferences. Furthermore, all successful applications of machine learning in hydrology are based on networks of fairly complex topology that require significant computational power and CPU time to train. For these reasons, the value of the standard hydrological models remains indisputable. In this study, we suggest employing machine learning models not as a substitute for hydrological models, but as an independent tool to assess their performance. We argue that this approach can help to unveil the anomalies in catchment data that do not fit in the employed hydrological model structure or configuration, and to deal with them without compromising the understanding of the underlying physical processes.

**Keywords:** machine learning; hydrological modelling; LSTM; recurrent neural networks; residual error modelling

## 1. Introduction

It is more than 50 years since the introduction of the multilayer perceptron model [1] whereas the first applications in hydrology started appearing almost 25 years ago including rainfall-runoff models [2] and short-term downstream flow forecasting [3]. More particularly, Minns and Hall [2] were among the first who applied recurrent neural networks (RNN) [4] in a hydrological application, stating that "*antecedent flow ordinates both perform the same function*" (note: distinguish between the rising limb and the recession) "*and provide additional information about the input pattern*". These early data-based models were fairly simple with at most two hidden layers and up to a dozen of hidden nodes, and seemed very promising in the dawn of the era of automatic data acquisition. A rainfall-runoff model could be created and calibrated based only on dynamic data (precipitation, evaporation, abstractions, etc.) obtained at high sampling frequencies from electronic sensors. However, these first machine learning (ML) models fell behind in performance compared with the standard hydrological models, conceptual or physically based.

The continuous increase of the computational power allowed more complex ML networks in a variety of hydrological applications especially in situations where the classical approaches are computationally demanding (for example, prediction of maximum flood

inundation [5], river flow prediction [6], water resources management [7], stochastic analysis [8], etc.). A milestone in time-series-related applications was the introduction of the long short-term memory (LSTM) [9]. LSTM units are used as hidden nodes in RNN and include, besides the input and output, a forget gate (see Figure 2 in [6]). This offers the advantage of assigning a dynamic state to each LSTM unit, which serves as a mechanism of memory. The topology of the LSTM networks is characterized by the number of different LSTM cells used in each time step (see Figure 1 in [6]) and the sequence length processed by the LSTM network at each step (see the figure in the section "LSTM Layer Architecture" in [10], which corresponds to the vertical direction of Figure 1 in [6]).

Machine learning with LSTM has outperformed the standard hydrological approach in many applications. For example, Ayzel et al. [11] demonstrated that an LSTM based model with 266,497 parameters achieved a higher generalization capacity than a parsimonious model for streamflow simulation in Barents, White and Baltic Seas. Kratzert et al. [12] introduced a modification to LSTM, the Entity-Aware-LSTM, which can take into account the catchment properties, and achieved better performance not only from the hydrological models that were calibrated regionally, but also from hydrological models that were calibrated for each basin individually. Lees et al. [13] applied LSTM in 669 catchments in Great Britain, which outperformed a suite of conceptual hydrological models.

This consistent performance superiority of ML in hydrological applications has been studied by Nearing et al. [14] who concluded that it is the constraints in the structure of the traditional models that prevents them from fully capturing the information in large-scale hydrological data sets. However, the efficiency of the LSTM models is not without a cost in computational complexity. As mentioned above, the model of Ayzel et al. [11] employed 266,497 parameters. Similarly, Lees et al. [13] have reported that it took 10 h to train an LSTM ensemble on a machine with 188 GB of RAM and a single NVIDIA V100 GPU.

An indirect approach to take advantage of ML in hydrological applications is to use it as a tool for pre-processing the data or post-processing the results of the standard hydrological models. For example, Iorgulescu and Beven [15] used ML to reveal anomalies in data sets, i.e., inconsistencies concerning the principal equations of standard models (e.g., water and energy balances). Solomatine et al. [16] have trained an ML model to serve as an estimator of the probability distribution of the output of a hydrological model. Althoff et al. [17] have suggested an elegant method to take advantage of the dropout technique (a common regularization strategy used in ML) to obtain ensemble predictions and quantify the uncertainty of hydrological models. Li et al. [18] have used a complex scheme that includes Box-Cox transformations, an LSTM network, and Bayesian inference to obtain probabilistic streamflow predictions. Aparicio et al. [19] have employed machine-learning models to estimate the instantaneous peak flow from the maximum mean daily obtained from the SWAT model. Noymanee et al. [20] have compared various alternative statistical and ML techniques for improving the flood forecasting efficiency of hydrological modeling. Yang et al. [21], have combined a physically-based distributed hydrological model with networks, computer vision, and a categorization approach.

In this study, inspired by the concepts found in the mentioned above works, we employ a simple ML network as a tool to assess the performance of hydrological models. A hydrologist always looks forward to improving the performance of his/her model. The simple ML network is used to assess a hydrological model and determine whether and how much it can be further improved. It should be noted that even if the optimization algorithm has achieved the best possible calibration of the hydrological model, the model may still not be able to achieve the best feasible fit because it is limited by its structural characteristics. However, as Beven suggests [22], "*If there are **consistent anomalies** between the conceptual structure of a hydrological model in a particular catchment and the nature of the hydrological processes in that catchment, then a deep learning (DL) model might well be able to capture that behaviour*". Therefore, by comparing the performance of a trained ML model with that of a calibrated hydrological model, the hydrologist can detect if there is room for further improvement and how to accomplish it by tuning the configuration and/or the

structure (e.g., spatial/temporal resolution, modules employed, parameters, etc.) of the hydrological model.

One could suggest using ML or deep learning approaches, like those in the previously mentioned studies, for benchmarking the performance of a hydrological model. However, this would require a significant amount of time, since, as previously mentioned, these models are notoriously CPU-intensive to train. On the other hand, the ML network employed in this study is minimalist. We opted for simplicity both to facilitate the applicability (an existing tool, like MATLAB ntstool, can be directly applied to the available data without any need for coding) and also to improve the generalization and reliability of this approach.

## 2. Materials and Methods

### 2.1. Hydrological Models

Two hydrological models were used for testing the assessment capacity of the suggested ML-based approach, LRHM and HYMOD2. LRHM [23] employs two simple model building blocks (direct runoff and soil moisture model) that are linearly combined to simulate the observed runoff (an idea related to the genetic programming model building [24]). HYMOD2 [25] is a conceptual parsimonious rainfall–runoff model. The schematic diagrams of these two models are displayed in Figure 1.

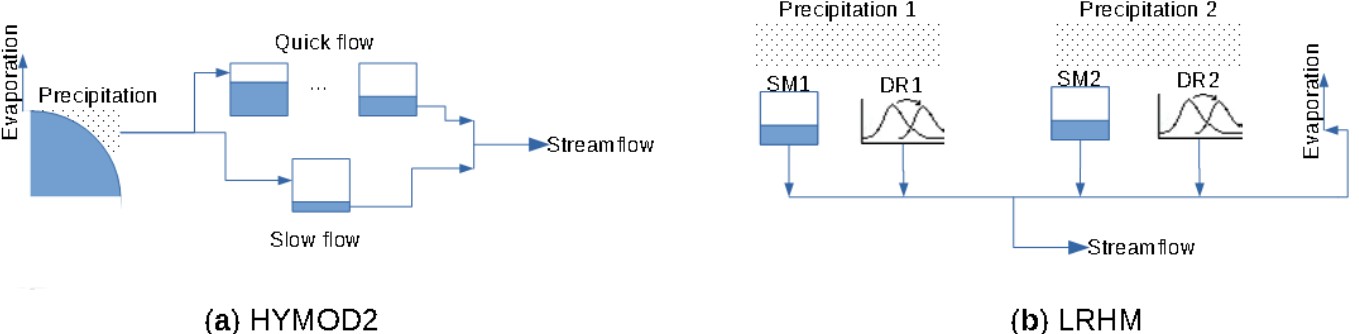

**Figure 1.** Schematic diagrams of HYMOD2 (**a**) and LRHM (**b**), SM1/2 soil moisture 1/2, DR1/2 direct runoff 1/2.

### 2.2. Case Studies

LRHM was applied in the case studies of Bakas, Alagonia, and Karveliotis presented in [23] (all in Nedon River, Greece). The catchment area of Nedon River is 118 km$^2$. The average annual precipitation depth is 1000 mm. The simulation time step was hourly with the observations extending from 1 September 2011 01:00 to 1 May 2014 00:00 (23,353 time steps). Time series of rainfall were obtained from 2 weather stations (distinct LRHM inputs), one for the higher altitudes of the catchment and one for the lower. Time series of potential evapotranspiration was obtained with the Penman-Monteith equation method.

In order to test the ability of the ML network to properly assess the performance of a hydrological model, the parameters of the LRHM in these three case studies were changed from what had been found as optimum values in [23]. More specifically, in the Bakas case study the coefficient of the multiple linear regression that corresponds to the first soil moisture module, which simulates the high flows, was set equal to 0, which results in the deactivation of this module; in the Alagonia case study, the interflow coefficient of the second soil moisture module, which simulates the lower flows was increased three orders of magnitude; in the Karveliotis case study, the coefficient of the multiple linear regression that corresponds to the first soil moisture module, which simulates the high flows, was set 4 times larger than the calibration value. This resulted in intentionally ill-calibrated models that exhibited typical hydrological model errors (underestimation/overestimation of high flows, failure to simulate flow intermittency, etc.) The results of the applications of the calibrated and ill-calibrated LRHM in the case studies of Bakas, Alagonia, and Karveliotis are shown in Figures 2–4.

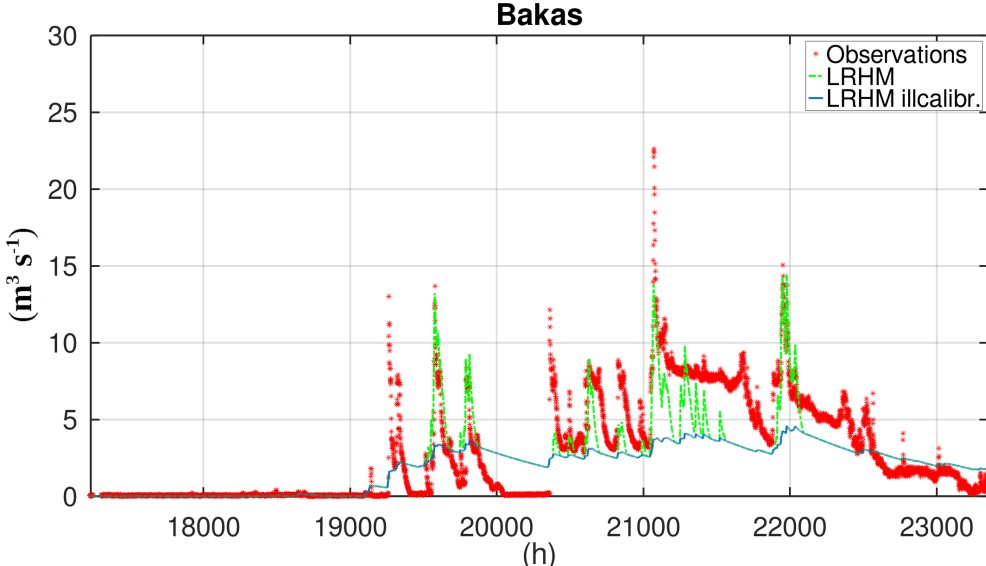

**Figure 2.** Application of LRHM in Bakas during its calibration period, corresponding intentionally ill-calibrated LRHM model and observations.

According to Figure 2, the ill-calibrated model LRHM simulates only the base flow and fails to simulate the high values of flows in the Bakas case study. According to Figure 3, the ill-calibrated model LRHM fails to reproduce the intermittency of flow (the minimum simulated flow value is not 0) in the Alagonia case study. According to Figure 4, the ill-calibrated model LRHM overestimates the high flows in the Karveliotis case study.

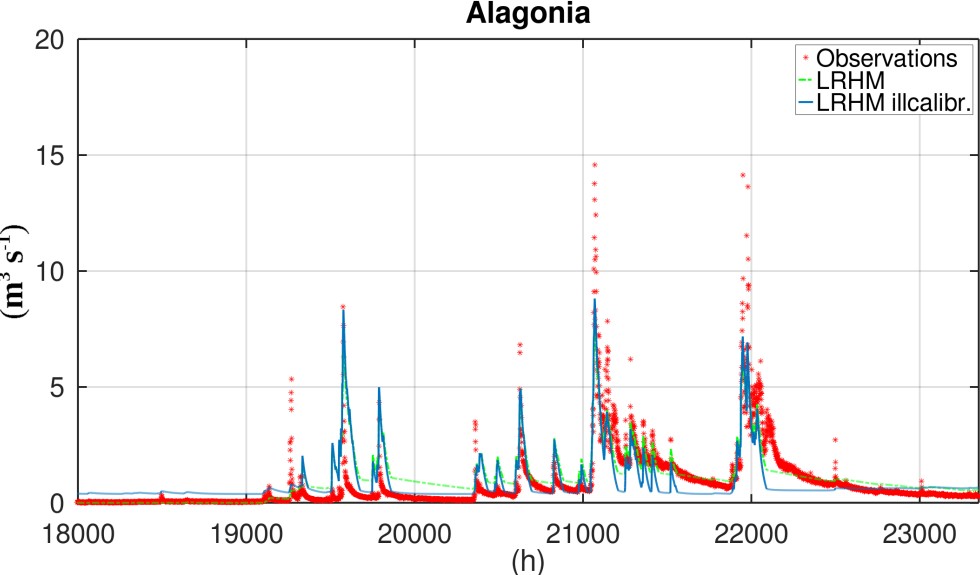

**Figure 3.** Application of LRHM in Alagonia during its calibration period, corresponding intentionally ill-calibrated LRHM model and observations.

In these three case studies, the training and test periods of the ML network coincide roughly (though this is not necessary) with the calibration and test periods of the hydrological model. As a general guideline, the training and test periods used in the ML network should ensure that all response patterns are represented evenly at both periods. For example, the ratio of the number of time steps with no flow to the steps with flow should be the same at both periods.

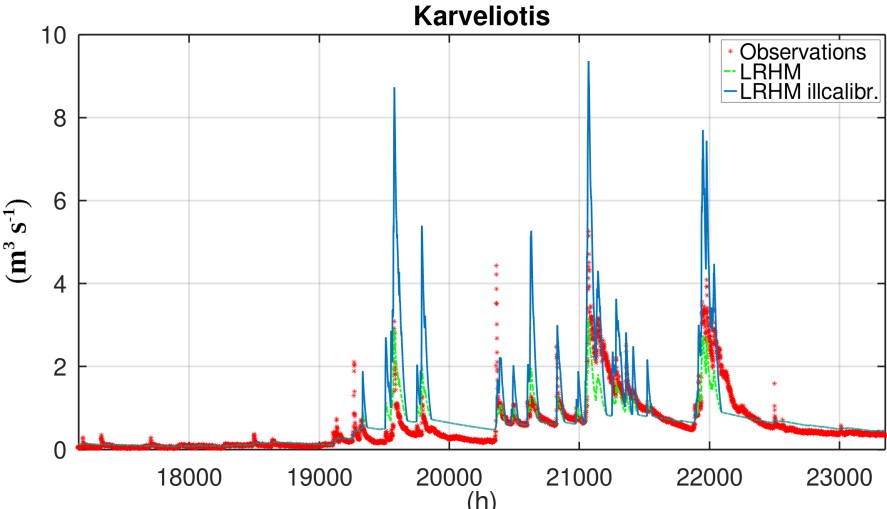

**Figure 4.** Application of LRHM in Karveliotis during its calibration, corresponding intentionally ill-calibrated LRHM model and observations.

In the previous case studies, the hydrological model was intentionally ill-calibrated to compromise the explanatory capacity of the model. To investigate the ability of the suggested methodology to detect performance limitations due to structural characteristics of a model, the suggested methodology was applied to the case study of Nyangores River, Kenya (obtained from [25]), employing two models, LRHM and HYMOD2. The catchment area of Nyangores River basin is 697 km$^2$. The average annual precipitation depth is 1500 mm. The simulation time step is daily. The potential evapotranspiration was obtained with the Hargreaves equation, whereas a single time series of precipitation was used, obtained with satellite-based estimations. The available data length is almost three and a half years, from 2007 to mid-2010.

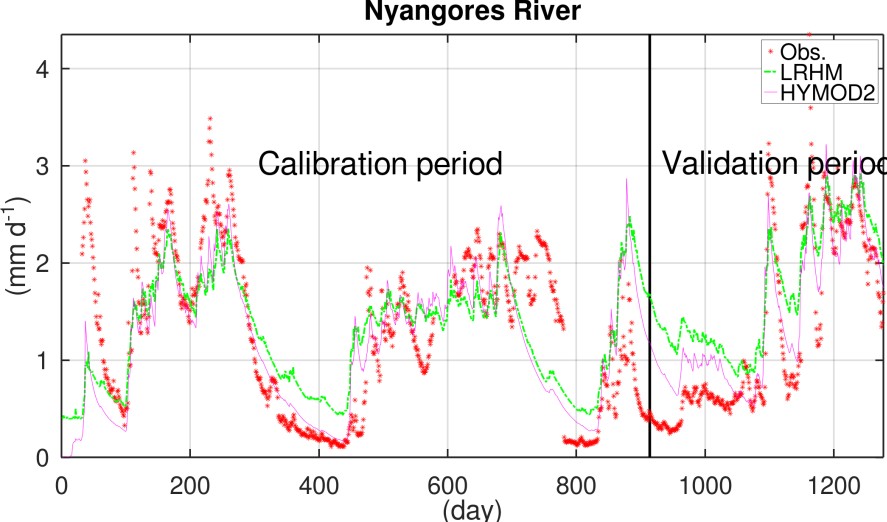

**Figure 5.** Application of LRHM and HYMOD2 in Nyangores River. Validation and calibration periods refer to the corresponding periods of the hydrological models.

The NSE coefficients achieved by LRHM and HYMOD2 during the calibration and test periods were 0.43 and 0.58, and 0.62 and 0.81 respectively [23]. These values indicate that HYMOD2 achieved better performance than LRHM in this case study (probably because the structure of HYMOD2 fits better the specific characteristics of this case study). Another thing that should be noticed is that both LRHM and HYMOD2 achieved better performance during the validation period than during the calibration period.

Figure 5 displays the simulation results of the two hydrological models, LRHM and HYMOD2, and the corresponding observations. The vertical axis corresponds to the specific discharge (streamflow divided by the catchment area, which is 697 km$^2$).

### 2.3. Data Shuffling

The significantly higher performance that the two hydrological models achieved during the validation period in Nyangores River case study indicates that the calibration and validation period exhibit different response patterns. For this reason, in this case study, the data for the training and test of the ML model was not obtained sequentially, as in the previous case studies. To preserve the basic statistical structure of the data, before splitting the data into the two sets, the data was fragmented into chunks of equal size. Then, each of the two sets was formed by concatenating, with random order, the data chunks (Figure 6). All data chunks are assigned to one of the two sets, and each chunk is assigned to only one set (training or test).

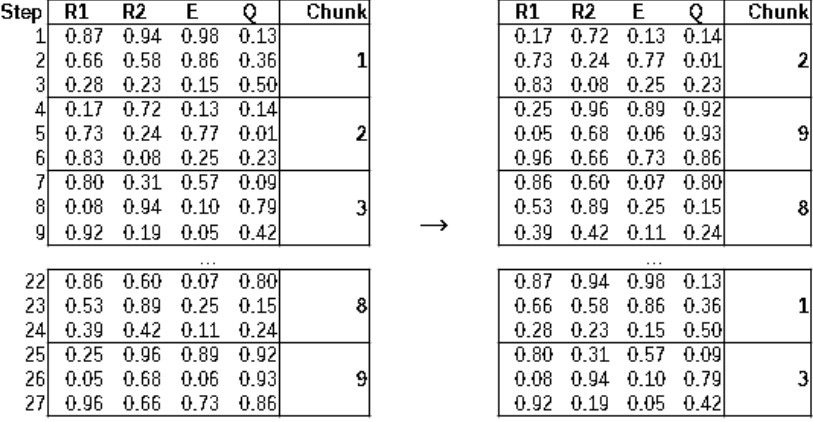

**Figure 6.** Example of shuffling data which include time series of rainfall (R1 and R2), evapotranspiration (E), and discharge (Q). R1, R2 and E exemplify the hydrological model inputs and can vary in number and type from a case study to another. The chunk size of this example is equal to 3 (much smaller than the actual size used in the case studies for the sake of the figure conciseness).

The concept of breaking the data into chunks and shuffling is similar to the concept of stochastic optimization, in which minibatches are employed [26]. The reason we didn't use stochastic minibatch (which is offered as a feature by the majority of the modern ML toolboxes) was that we needed to have control over the shuffled data set in order to be able to obtain the corresponding objective function value of the hydrological model. According to Wilson and Martinez [27] smaller minibatch sizes improve the regularization. However, the smaller the minibatch size gets, the denser the temporal discontinuities of the input time series, hence, the more blurred their statistical structure (compared to the structure of the original, sequential, data). Therefore, there is an optimal minibatch size (we prefer the term 'chunk' over the term 'minibatch' because we employ batch training, i.e., the whole data set is processed at one step during each epoch), which should be used. After a preliminary study, this optimal size was found to be close to 10 times the sequence length of the employed ML network approximator.

After training the ML network a single time, it was applied multiple times on data sets formed from the original set with shuffling. The corresponding values of the loss function were recorded and their mean and variance were calculated. This technique is similar to cross-validation, which is frequently used in ML applications as an estimator of generalization performance. According to Cawley and Talbot [28], a low variance of the loss function of the ML is equally important with a low average value in avoiding overfitting.

### 2.4. Performance Assessment

According to Smith et al. [29] the relationship between hydrological model residuals and observations at a specific time step can be represented with the following formula:

$$y_\mathrm{o} = y_\mathrm{m}(z; \theta) + y_\mathrm{e} \tag{1}$$

where $y_\mathrm{o}$ is the observed value, $y_\mathrm{m}(z; \theta)$ is the model output, $\theta$ is the vector with the model parameters, $z$ is the vector with the dynamic input data, and $y_\mathrm{e}$ is the residual.

If we denote with $f$ the function that describes the consistent anomalies between the observations and model results, then the residual can be written as:

$$y_\mathrm{e} = f(x; \theta) + \epsilon \tag{2}$$

where $x$ is a vector including a combination of the dynamic input data $z$ and the model results $y_\mathrm{m}(z; \theta)$, and $\epsilon$ is the uncorrelated, homoscedastic, and zero-inflated model error.

Equations (1) and (2) can be combined into a single equation:

$$y_\mathrm{o} = (y_\mathrm{m}(z; \theta) + f(x; \theta)) + \epsilon \tag{3}$$

The definition of the best feasible model $\Phi(x)$ achievable with the available data can be obtained from Equation (3) (note that $z \subset x$):

$$\Phi(x) = y_\mathrm{m}(z; \arg\min_{\theta} |y_\mathrm{e}|) + f(x; \arg\min_{\theta} |y_\mathrm{e}|) \tag{4}$$

$$\Phi(x) = y_\mathrm{o} - \epsilon \tag{5}$$

Equation (4) describes how the best feasible model can be obtained from the assessed hydrological model, and Equation (5) gives the definition of the best feasible model, a model of which the residual is uncorrelated, homoscedastic, and zero-inflated.

An ML model can be employed to obtain the approximation $\widehat{\Phi}(x)$ of the best model $\Phi(x)$ by fitting the ML network to $y_\mathrm{o} - \epsilon$, or, since $\epsilon$ is uncorrelated, directly to $y_\mathrm{o}$. Substituting $\widehat{\Phi}(x)$ in Equation (5) and solving for the error $\hat{\epsilon}$ it is obtained that:

$$\hat{\epsilon} = y_\mathrm{o} - \widehat{\Phi}(x) \tag{6}$$

If $\hat{\epsilon}$ is similar to $y_\mathrm{e}$, then from Equation (1) it can be inferred that:

$$\widehat{\Phi}(x) \approx y_\mathrm{m}(z; \arg\min_{\theta} |y_\mathrm{e}|) \tag{7}$$

This means that the approximator of the best feasible model coincides with the assessed model. If Equation (7) is substituted into Equation (4), it is obtained that:

$$f(x; \arg\min_{\theta} |y_\mathrm{e}|) \to 0 \tag{8}$$

which means that if $\hat{\epsilon} \approx y_\mathrm{e}$, the assessed model does not exhibit consistent anomalies (the model error is uncorrelated, homoscedastic, and zero-inflated), i.e., the deterministic relationship between inputs and outputs is fully described, therefore no further improvement of performance is achievable. On the contrary, if $\hat{\epsilon}$ is lower than $y_\mathrm{e}$, i.e., the ML approximator accomplishes a better performance than the hydrological model, then there is some information in the data which the structure or setup of the hydrological model is not taking into account.

To increase the confidence in the results, two alternative methods were used to prepare the ML network serving as the approximator $\widehat{\Phi}(x)$. In the first method, Cortexsys (a deep learning toolbox for MATLAB and GNU Octave) was employed to prepare a standard RNN. In the second method, Cortexsys was employed to prepare a recurrent network

with LSTM cells. Both ML networks were trained with the gradient descent algorithm ADADELTA [30]. A z-score normalization was used for input data (minmax normalization was also tested without any benefit), no minibach or dropout techniques were used. The network topology is shown in Figure 7.

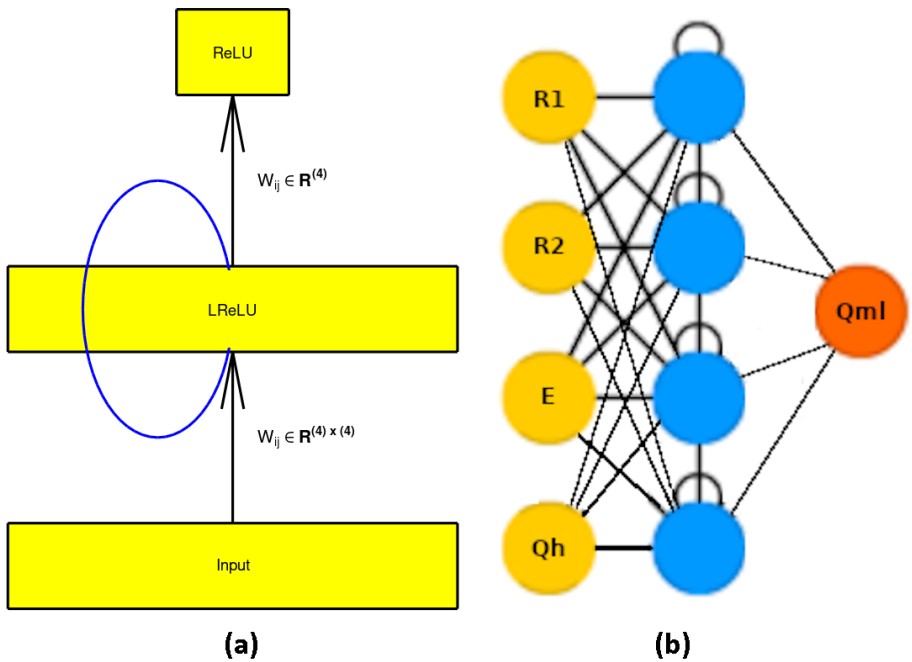

**Figure 7.** Example of an ML approximator $\widehat{\Phi}(x)$ of the best feasible model $\Phi(x)$. Left panel (**a**) displays the activation functions (LReLU and ReLu), the dimensions of the weight matrices and the recurrent (indicated with a blue ellipse) hidden layer. Right panel (**b**) displays the inputs and outputs of the employed ML approximator. The assessed hydrological model has three inputs (precipitations R1 and R2, and evapotranspiration E) and one output (Qh). These are the four inputs of the ML approximator. If the ML approximator output, Qml, is closer to the observations than the hydrological model output, Qh, then there is some information in the data that the structure or setup of the hydrological model is not taking into account.

The suggested number of hidden nodes/cells is equal to the number of ML inputs. This means 4 hidden nodes/cells for the Bakas, Alagonia, and Karveliotis case studies and 3 nodes/cells for the Nyangores River case study. The activation functions were LReLU for the hidden, and ReLU for the output layers [31]. The total number of parameters for the standard RNN with four inputs was $4 \times 4 + 4$ weights and $4 + 1$ biases. The sequence length is a hyperparameter that should be tuned individually for each case study. For the Bakas, Alagonia, and Karveliotis case studies the sequence length was 4 whereas for the Nyangores River case study was 2. The ML model inputs were the dynamic inputs of the hydrological model, i.e., precipitation (2 time series for Bakas, Alagonia, and Karveliotis, and 1 for Nyangores River) and evapotranspiration, and the simulated flow by the hydrological model. The loss function was the mean squared error (MSE). This was also the objective function employed in the hydrological models.

## 3. Results

### 3.1. Example of Ill-Calibrated Model—Failing in High Flows

Figure 8 displays the simulated discharge by the intentionally ill-calibrated hydrological model LRHM and the approximation of the best model that can be obtained with the available data in the Bakas case study (Figure 8a,b corresponds to LSTM and RNN). According to this figure, a significantly better model can be achieved with the available data, especially concerning the simulation of the higher flows.

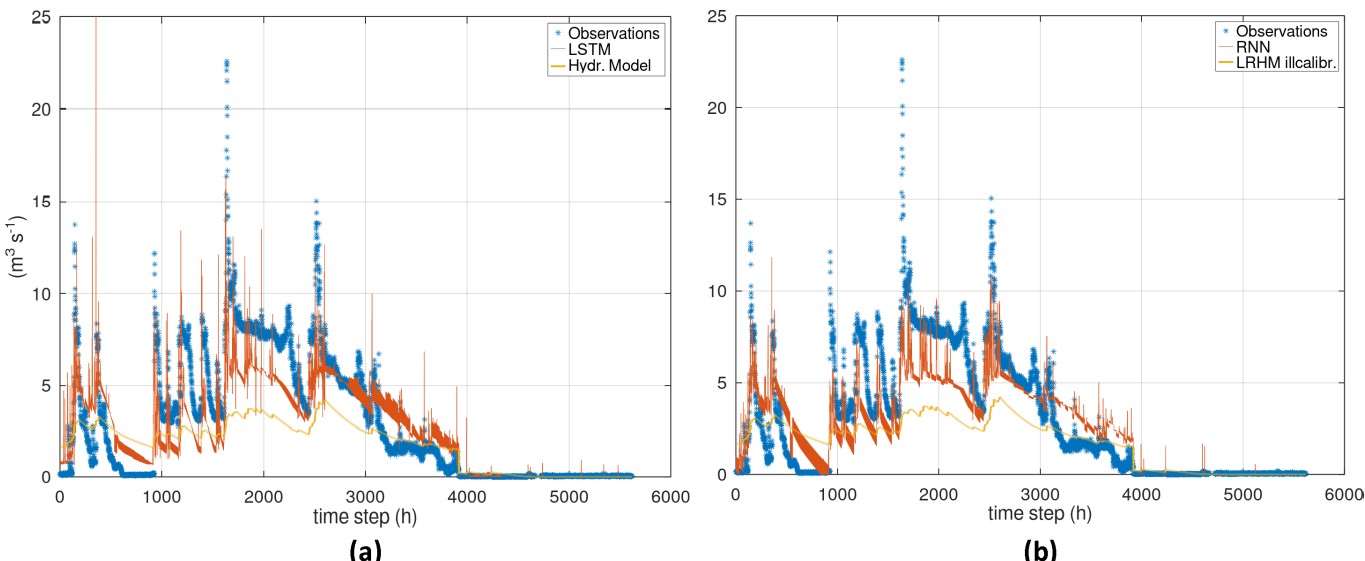

**Figure 8.** Assessment of the ill-calibrated hydrological model LRHM in Bakas. LSTM (**a**) and RNN (**b**) employed to approximate the best model that can be obtained with the available data during the test period.

Table 1 displays the improvement of the performance of the ML approximators over the calibrated and ill-calibrated hydrological model LRHM in the Bakas case study. Both ML approximators (LSTM and RNN networks) achieved significant improvement over the ill-calibrated model both in the training and test, which indicates, as expected, that the ill-calibrated hydrological model does not capture well the relationship between the model inputs and output. Furthermore, the ML approximators achieved a slightly better performance than the calibrated model, which indicates that the strength (or the complexity) of the hydrological model LRHM marginally suffice to describe the hydrological response at the location of Bakas.

**Table 1.** Percentage improvement of the performance of the ML approximator over the calibrated and ill-calibrated LRHM hydrological model in the Bakas case study.

|  | Train | | Test | |
|---|---|---|---|---|
|  | **LSTM** | **RNN** | **LSTM** | **RNN** |
| Calibrated LRHM MSE | 1.87 | | 3.59 | |
| ML approximator MSE | 1.22 | 1.29 | 3.45 | 3.10 |
| Improvement | 35% | 31% | 4% | 14% |
| Ill-calibrated LRHM MSE | 5.34 | | 7.18 | |
| ML approximator MSE | 2.76 | 3.25 | 4.89 | 3.61 |
| Improvement | 48% | 39% | 32% | 50% |

### 3.2. Example of Ill-Calibrated Model—Overestimating Low Flows

Figure 9 displays the simulated discharge by the intentionally ill-calibrated hydrological model LRHM and the approximation of the best model that can be obtained with the available data in the Alagonia case study.

Comparing the results of the ML approximator (Figure 9a,b corresponds to LSTM and RNN) with that of the intentionally ill-calibrated hydrological model, it can be inferred that the latter introduces a bias during low-flow periods (non-zero-inflated residual) and more steep recession curves.

Table 2 displays the improvement of the performance of the ML approximators over the calibrated and ill-calibrated hydrological model LRHM in the Alagonia case study. Both ML approximators (LSTM and RNN) achieved significant improvement of the ill-calibrated model both in the calibration and the test period, as expected. The ML approximators

achieved a rather better performance than the calibrated model also, which indicates that the strength (or the complexity) of the hydrological model marginally suffice to describe the hydrological response at the location of Alagonia.

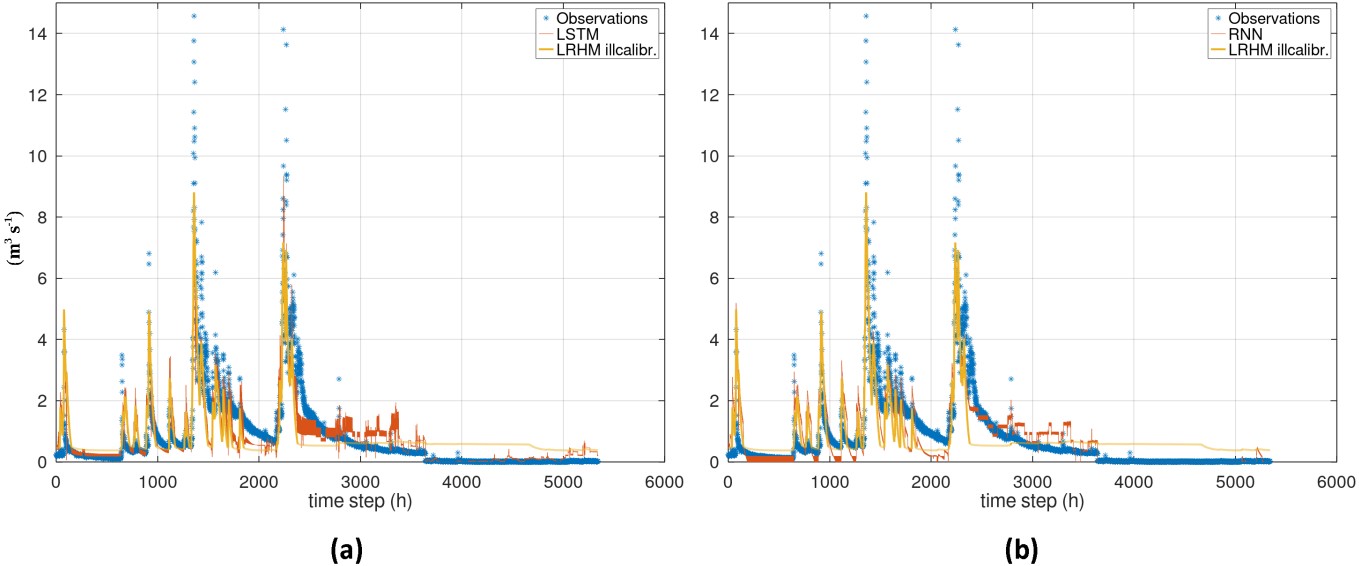

**(a)**                                    **(b)**

**Figure 9.** Assessment of the ill-calibrated hydrological model LRHM in Alagonia. LSTM (**a**) and RNN (**b**) employed to approximate the best model that can be obtained with the available data during the test period.

**Table 2.** Percentage improvement of the performance of the ML approximator over the calibrated and ill-calibrated LRHM hydrological model in the Alagonia case study.

|  | **Train** | | **Test** | |
|---|---|---|---|---|
|  | **LSTM** | **RNN** | **LSTM** | **RNN** |
| Calibrated LRHM MSE | 0.37 | | 0.42 | |
| ML approximator MSE | 0.33 | 0.33 | 0.38 | 0.38 |
| Improvement | 11% | 10% | 9% | 9% |
| Ill-calibrated LRHM MSE | 0.72 | | 0.59 | |
| ML approximator MSE | 0.48 | 0.53 | 0.50 | 0.46 |
| Improvement | 33% | 26% | 15% | 22% |

### 3.3. Example of Ill-Calibrated Model—Overestimating High Flows

Figure 10 displays the simulated discharge by the intentionally ill-calibrated hydrological model LRHM and the approximation of the best model that can be obtained with the available data in the Karveliotis case study (Figure 10a,b corresponds to LSTM and RNN). Comparing the results of the ML approximator with that of the intentionally ill-calibrated hydrological model, it can be inferred that the latter introduces overestimates at the high flows.

Table 3 displays the improvement of the performance of the ML approximators over the calibrated and ill-calibrated hydrological model LRHM in the Karveliotis case study. Both ML approximators (LSTM and RNN) achieved a huge improvement over the ill-calibrated model both in the calibration and the test period, which indicates that the ill-calibrated hydrological model falls far behind the best achievable model. The ML approximators achieved equivalent performance with the calibrated model, which indicates that the calibrated hydrological model LRHM has, most probably, achieved the best possible performance with the available data.

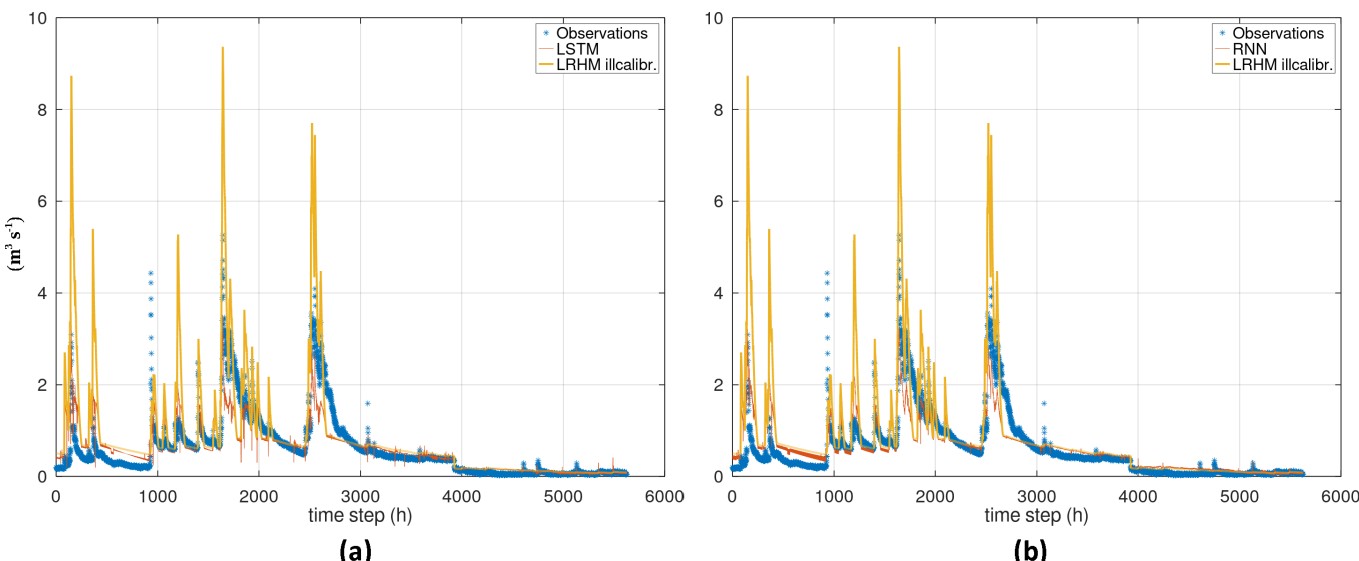

**Figure 10.** Assessment of the ill-calibrated hydrological model LRHM in Karveliotis. LSTM (**a**) and RNN (**b**) employed to approximate the best model that can be obtained with the available data employed during the test period.

**Table 3.** Percentage improvement of the performance of the ML approximator over the calibrated and ill-calibrated LRHM hydrological model in the Karveliotis case study.

| | **Train** | | **Test** | |
| --- | --- | --- | --- | --- |
| | **LSTM** | **RNN** | **LSTM** | **RNN** |
| Calibrated LRHM MSE | 0.038 | | 0.140 | |
| ML approximator MSE | 0.036 | 0.038 | 0.136 | 0.137 |
| Improvement | 7% | 0% | 3% | 2% |
| Ill-calibrated LRHM MSE | 1.008 | | 0.672 | |
| ML approximator MSE | 0.037 | 0.039 | 0.157 | 0.149 |
| Improvement | 96% | 96% | 77% | 78% |

### 3.4. Example of Structural Insufficiency

In the Nyangores River case study, two hydrological models were assessed, the model LRHM, and the model HYMOD2. Both models were assessed with the optimum parameters obtained after their calibration (see [23,25]). HYMOD2 achieved better performance than LRHM. Therefore, it is expected from the ML approximator to detect that the LRHM performance is not the best that can be obtained.

The ML approximator in this case study was trained and validated with the shuffling approach. To accomplish this, the ML approximators of the two hydrological models were trained once, but applied multiple times, each time with a different data set obtained after shuffling the original data set. The single data set used for training was selected to have a similar loss function value of the ML network in the training and test. The mean value of the loss function and its variance (LSTM and RNN were employed, like in the previous case studies, but their results were averaged) were compared against the corresponding values of the hydrological models (LRHM and HYMOD2).

It should be noted that the hydrological models were not run multiple times, like the ML approximators, only their outputs were shuffled and compared against the corresponding observations to obtain performance values (the hydrological models are constrained by the continuity equations; shuffling the input data and applying the hydrological models would create unrealistic artefacts, e.g., recession curves at every time-stamp jump due to the shuffling). The statistics of these performance values are compared against the statistics of the corresponding ML approximators (see Tables 4 and 5).

**Table 4.** Mean value and variance of the ML approximator loss function and LRHM performance metric in the Nyangores River case study.

| | ML Approximator | | LRHM | |
|---|---|---|---|---|
| | Train | Test | Train | Test |
| Mean | 0.33 | 0.33 | 0.37 | 0.38 |
| Variance | 0.001 | 0.011 | 0.002 | 0.020 |

**Table 5.** Mean value and variance of the ML approximator loss function and HYMOD2 performance metric in the Nyangores River case study.

| | ML Approximator | | HYMOD2 | |
|---|---|---|---|---|
| | Train | Test | Train | Test |
| Mean | 0.26 | 0.28 | 0.27 | 0.29 |
| Variance | 0.001 | 0.008 | 0.001 | 0.010 |

Comparing the values of Tables 4 and 5, it can be obtained the percentage reductions of the mean value and variance achieved by the ML approximator over the hydrological models LRHM and HYMOD2. These values are presented in Table 6. According to this table, the improvement the ML approximator achieved over LRHM is much greater than the improvement over HYMOD2, as was expected. However, Table 6 indicates that a model even better than HYMOD2 is achievable. In addition, the lower variance of the ML approximator indicates that this anticipated better hydrological model will also regularize better.

**Table 6.** Percentage reduction of the mean value and variance of the ML approximator over the hydrological models LRHM and HYMOD2.

| | LRHM | | HYMOD2 | |
|---|---|---|---|---|
| | Train | Test | Train | Test |
| Mean | 8% | 10% | 6% | 5% |
| Variance | 36% | 28% | 21% | 27% |

Finally, it is noted that the direct application of the ML approximator, as it was employed in the previous case studies (Bakas, Alagonia, and Karveliotis), did not achieve improvement over the hydrological models, as it was hampered by the much better performance of the hydrological models during the test period. However, it was found that this difficulty could be circumvented if the ML approximators were trained with shuffled data sets of which the hydrological models' performance during the training and test were equivalent. Then, these ML approximators can be applied to the original, sequential, data and give a more reliable assessment than the direct application. Table 7 displays this improvement of the performance of the ML approximators, trained with shuffled data sets but applied with the original, sequential data, over the LRHM and HYMOD2 hydrological models in the Nyangores River case study.

**Table 7.** Percentage improvement of the performance of the ML approximators over the LRHM and HYMOD2 hydrological models in the Nyangores River case study (sequential data).

| | Train | | Test | |
|---|---|---|---|---|
| | LSTM | RNN | LSTM | RNN |
| LRHM | 11% | 3% | 16% | 3% |
| HYMOD2 | 3% | 8% | 5% | −10% |

Figure 11 displays the simulated discharge by the hydrological model LRHM and the approximation of the best model that can be obtained with the available data (Figure 11a,b

corresponds to LSTM and RNN). Comparing the results of the LSTM ML approximator with that of LRHM hydrological model, it can be inferred that the former, as it is indicated also in Table 7, achieved better performance.

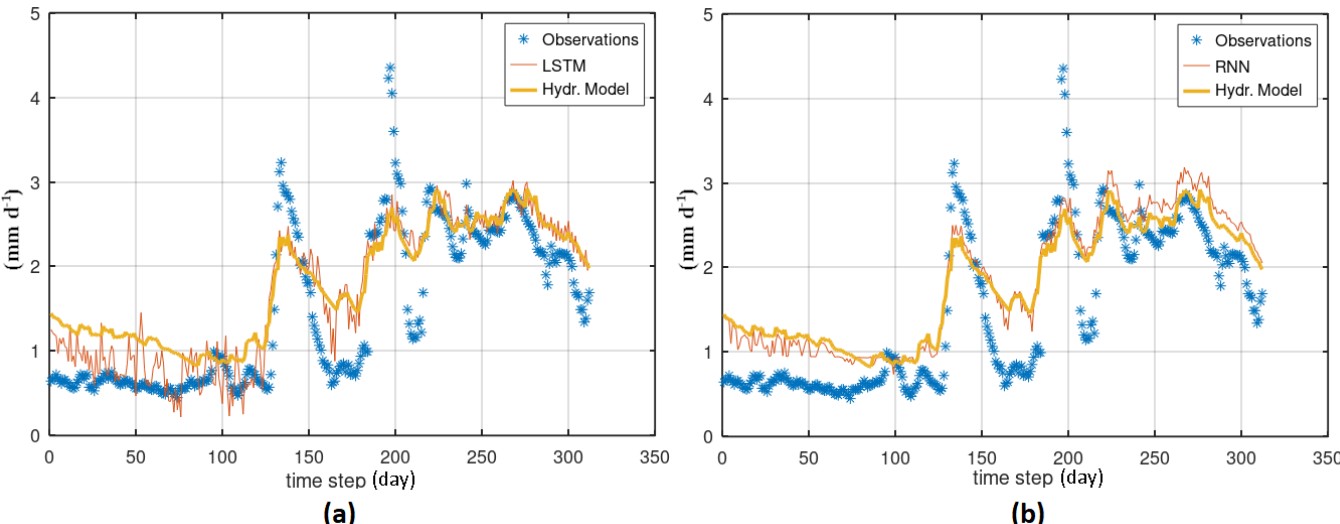

**Figure 11.** Assessment of the LRHM model in the Nyangores River. LSTM (**a**) and RNN (**b**) employed to approximate the best model that can be obtained with the available data during the LRHM test period.

Figure 12 displays the simulated discharge by the hydrological model HYMOD2 and the approximation of the best model that can be obtained with the available data (Figure 12a,b corresponds to LSTM and RNN). Comparing the results of the ML approximators with that of the hydrological model HYMOD2, it can be inferred that it did not achieve a better performance.

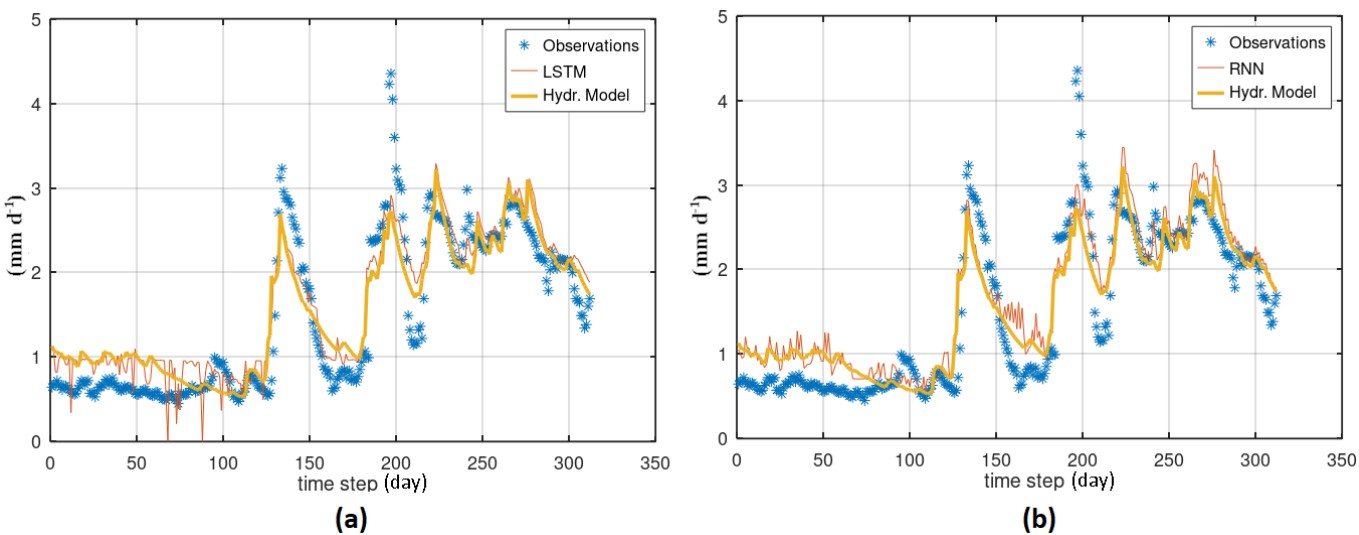

**Figure 12.** Assessment of the HYMOD2 model in the Nyangores River. LSTM (**a**) and RNN (**b**) employed to approximate the best model that can be obtained with the available data during the HYMOD2 test period.

## 4. Discussion

On closer inspection, the outputs of the ML approximators exhibit some unrealistic characteristics (e.g., large spikes, unjustified oscillations, etc.). It should be noted that similar behaviour is typical in plain RNNs. For example, Bao et al. compared various ML

models to predict stock indexes. They found that LSTM and RNN have large variations and distances to the actual data (see Figure 8 in [32]). To improve the performance, they combined an LSTM network with wavelet transforms (WT) and stacked autoencoders. Other researchers have reported similar behaviour while employing RNN to perform sensor fusion for an inertial measurement unit [33]. The suggested solution was to use Fourier Transformation and remove high-frequency signal components. As was mentioned in the introduction section, ML can achieve superior performance in hydrological applications, but only after paying the price of increased computational complexity. However, the ML approximator is not intended for hydrological simulations, but for the assessment of hydrological models. To make the assessment procedure practical, the ML approximator was designed to be as simple as possible, hence the oscillations and spikes. Nevertheless, these artefacts should not be a source of concern. If the ML approximator achieves better performance, despite any high-frequency oscillations (but not because of), then a better hydrological model is achievable.

The comparison of a hydrological model against the simulation of the ML approximator can give clear indications of the performance sufferings that can be addressed. Similar conclusions cannot be obtained after a comparison of the model directly with the observations because it cannot be guaranteed that improving an obvious failure of the model to reproduce a specific characteristic of the response will not deteriorate the reproduction of another characteristic. For example, this comparison in Figure 8 (ill-calibrated LRHM against RNN and LSTM) indicates that a better model in Bakas can be obtained to simulate more accurately both the high and low flows; on the other hand, the comparison in Figure 9 indicates that only the performance during the low-flow periods can be improved.

In the case study of Nyangores River, the significant difference in the performance of the hydrological models between calibration and validation periods made difficult the direct approach employed in the other case studies. The shuffling approach, inspired by the cross-validation technique, allowed the study of the variance of the loss function of the ML approximator and the variance of the performance metric of the hydrological model. In the Nyangores River case study, the ML approximator demonstrated lower variance for both assessed hydrological models, LRHM and HYMOD2. To further study the importance of the variance, the shuffling approach was tried in the Karveliotis case study, at a second round, this time only for the calibrated LRHM, for which the direct application of the ML approximator achieved only a marginal performance improvement. The variance of the ML approximator, obtained after the shuffling and the repeated applications, was slightly larger than the variance of the calibrated LRHM suggesting that the performance improvement, in this case, is not only marginal but also does not generalize, which increases the credibility of the indication obtained with the direct application (see Section 3.3) that a better model performance than that already achieved by LRHM is not feasible.

The shuffling of the data, though introducing some additional workload, offered a twofold benefit in the application of the ML approximator. First, it facilitated the ML model training in situations where the training and test periods presented different response patterns. Second, it allowed the evaluation of the variance of the loss function, as was mentioned in the previous paragraph. This beneficial technique is not applicable in hydrological models because it violates the continuity equations, the cornerstone of these models. Shuffling carefully the data (to preserve the important correlation characteristics) offers an important advantage to ML models over standard approaches, which should be considered in hydrological applications.

As mentioned previously, the ML approximator helps to identify whether there is some information in the data that the structure or setup of the hydrological model is not taking into account. In most cases, this happens because of some deficit or weakness of the hydrological model. An interesting situation is when the observations are obtained by a sensor that introduces systematic error. In this case, the ML approximator will yield a better fit than the hydrological model, which is restricted by the water balance equation. Yet, in this case, the hydrological model predictions will be closer to reality. Though modern

sensors are reliable and tested in laboratories to ensure no systematic errors, the possibility of such an untoward event cannot be excluded. At this point, we need to acknowledge the agnostic nature of any kind of model. If the observations are found at some point to suffer from systematic errors, the model should be run again with the new, closer to reality, data. If no error can be proven in the available data, then the reality for the model is the data.

Finally, it is worth noticing that the ML approximator $\widehat{\Phi}(x)$ did achieve better performance than the ill-calibrated models in Nedon and the less efficient hydrological model in the Nyangores River case study, indicating the potentials to improve the model performance. However, this performance improvement did not exceed the performance of the corresponding calibrated models and the more efficient hydrological model (e.g., compare the values of the ML approximator in Table 4 and HYMOD2 in Table 5). This means that the ML approximator $\widehat{\Phi}(x)$ actually gives a better model than $y_m(z; \theta)$, but not the best feasible model $\Phi(x)$. Apart from what has been mentioned in the beginning of this section, another reason is the assumption that there is a function $f(x; \theta)$ such that $\epsilon$ in Equation (2) is uncorrelated, homoscedastic, and zero-inflated. Since the complexity of $f(x; \theta)$ is equivalent to the complexity of the approximator $\widehat{\Phi}(x)$, which in the proposed methodology is kept minimal, the aforementioned assumption may not be feasible for every assessed model structure and configuration. The alternative to bypass this restriction would be to allow a function $f(x; \theta)$ of arbitrary complexity, which in the end would result in ML applications similar to that appearing in various recent publications. Consequently, any advantage regarding the time required to train (the time required on an AMD EPYC dual-core processor was less than 2 minutes for LSTM and less than 1 minute for RNN) and to apply the ML approximators would be lost.

## 5. Conclusions

In this study, we developed an approach to help a hydrologist to answer the question "Can my model perform any better with the available data?". To answer this question, we suggest employing a simple machine learning network that can be easily prepared and trained. The network inputs are the inputs and outputs of the hydrological model. If the machine learning model achieves better performance, then there is some information in the data that the structure or setup of the hydrological model is not taking into account.

The proposed methodology can be applied with simple ML tools that are straightforward and require no coding or data curation. However, this direct simple approach is reliable only when the assessed hydrological model performs almost equivalently during the training and test. A large difference in performance between these two periods indicates that important response patterns are not evenly represented in the data sets of the training and test. In these cases, a more sophisticated approach that includes data curation (shuffling carefully to preserve the statistical structure) is required to have a reliable model assessment.

When assessing a model with data shuffling (a method similar to cross-validation) the variance of the loss function is a metric of how well the model generalizes. If the machine learning achieves lower variance than the hydrological model, this is an additional indication that a better model can be prepared with the available hydrological data. This was confirmed by the application of the shuffling method into two case studies (Nyangores River and Karveliotis). In the former case study, the mean loss function and its variance of the machine learning model were lower than that of the hydrological model, indicating the potentials of a better model, whereas in the latter the mean loss function was marginally lower but the variance was higher, indicating that the best feasible model has already been achieved.

The suggested methodology could be used as a filter to improve the efficiency of hydrological models. However, as mentioned previously, it is not guaranteed that the improved performance after the filtering is the best that can be obtained with the available data. For this reason, it is recommended to be used principally as an assessment tool for crafting hydrological models, either by improving the calibration or by adding/modifying

physical/conceptual assumptions to the model. Finally, it should be noticed that the suggested methodology is generic. It can be used with any model, e.g., hydraulic, even a financial model, to evaluate the capacity of the assessed model to fully describe the deterministic relationship between the model inputs and outputs.

**Author Contributions:** Conceptualization, E.R.; methodology, E.R.; software, E.R. and P.D.; validation, P.D. and V.B.; formal analysis, E.R.; investigation, E.R.; resources, V.B.; data curation, E.R.; writing—original draft preparation, E.R.; writing—review and editing, P.D. and V.B.; visualization, E.R.; supervision, E.R.; project administration, E.R. All authors have read and agreed to the published version of the manuscript.

**Funding:** This research received no external funding.

**Institutional Review Board Statement:** Not applicable.

**Informed Consent Statement:** Not applicable.

**Data Availability Statement:** Data supporting reported results can be found in [23] (Bakas, Alagonia, and Karveliotis case studies) and in [25] (Nyangores River case study).

**Acknowledgments:** This work was supported by the Inner Scholarship of National Observatory of Athens "Low computational burden flood modelling in small to medium-sized water basins in Greece".

**Conflicts of Interest:** The authors declare no conflict of interest.

## Abbreviations

The following abbreviations are used in this manuscript:

| | |
|---|---|
| ML | Machine learning |
| MSE | Mean squared error |
| NSE | Nash-Sutcliffe efficiency |
| LSTM | Long short-term memory |
| RNN | Recurrent neural network |

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
