# Peer review of "Machine Learning in Assessing the Performance of Hydrological Models"

_hydrology, doi:10.3390/hydrology9010005_

Round 1

Reviewer 1 Report

The authors propose to use ML model results to examine the physical model results. If the physical model results were worse than the ML model, then there must be some flaw in the physical model and the physical model can be improved. It is a very interesting idea, but I'm not convinced by the methods and results in the current manuscript. 

While the author claimed, “Similar conclusions cannot be obtained after a comparison of the model directly with the observations because it cannot be guaranteed that improving an obvious failure of the model to reproduce a specific characteristic of the response will not deteriorate the reproduction of another characteristic.”  using their two cases (Line 319~314). I would argue a more sophisticated ML may eventually outperform any physical model, as long as the physical model doesn't fully match the observation data.

I'd like to challenge the authors with a more specific question. Assuming the sensors have a systematic drift and produce biased observation. A physical model will "fail" to match the biased data,  but it is highly likely the ML model will still match the observation well. Using the approach the author suggested will be misleading that the physical model needs to be improved.
